# Exploring the prevalence of atopic disease among adult cancer survivors: insights from the 2021 NHIS

**Lu Xiao[1]⚬, Yuhui Shen[2]⚬, Nana Xiang[3], Juebo Yu⊙[1]\***

1 Department of Otolaryngology, Affiliated Hospital of Yangzhou University, Yangzhou University, Yangzhou, Jiangsu, China, 2 Department of Blood Donation Service, Yangzhou Central Blood Station, Yangzhou, Jiangsu, China, 3 Department of Otolaryngology, Kunshan Hospital Integrated Traditional Chinese and Western Medicine, Kunshan,Suzhou,Jiangsu, China

⚬ These authors contributed equally to this work.
\* yujuebo2004@163.com

## Abstract

Cancer survivors who undergo radiotherapy, immunotherapy, and chemotherapy experience a decline in immune system function. Atopic diseases (ADs) are associated with systemic immunosuppressant medications and exposure to environmental allergens. However, the relationship between cancer survivorship and the prevalence of ADs remains poorly understood. This study investigated whether the prevalence of ADs is increased among cancer survivors. A cross-sectional survey was conducted using data from the 2021 US National Health Interview Survey (NHIS). The data were on 28,617 adults aged 18 years and older. Data analysis was performed between May 11, 2024, and July 30, 2024. The NHIS participants reported whether they had a history of current hay fever, still asthma, current skin allergy, and/or current food allergy. A history of cancer was defined on the basis of self-reported data from the NHIS. The weighted prevalence and 95% confidence intervals (CIs) of ADs were calculated. Comparisons of AD prevalence between cancer survivors and the general population were conducted using chi-square tests and multiadjusted logistic regression models. Of the 28,617 adults sampled in the 2021 NHIS, the prevalence of ADs among the cancer survivors was as follows: current hay fever (54.5%; 95% CI, 52.6%–56.4%), persistent asthma (14.7%; 95% CI, 12.9%–13.9%), current skin allergy (13.6%; 95% CI, 10.1%–15.0%), and current food allergy (11.2%; 95% CI, 10.0%–12.4%). In multivariable logistic regression models adjusted for age, sex, race/ethnicity, family income, educational level, insurance status, body mass index, general health status, region, location, and comorbidity burden, the cancer survivors had significantly higher odds of ADs than did the general population. In particular, they had higher odds of current hay fever (aOR, 1.14; 95% CI, 1.05–1.23), persistent asthma (aOR, 1.17; 95% CI, 1.03–1.33), current skin allergy (aOR, 1.26; 95% CI, 1.13–1.41), and current food allergy (aOR, 1.22; 95% CI, 1.08–1.38). The findings

**Data availability statement:** Data and code are held in Dryad, with the DOI https://doi.org/10.5061/dryad.2280gb63m.

**Funding:** The author(s) received no specific funding for this work.

**Competing interests:** The authors have declared that no competing interests exist.

of this cross-sectional study indicated that the prevalence of ADs was significantly higher in cancer survivors than in the general population. Future research is required to elucidate the underlying mechanisms, optimize treatment strategies, and enhance the overall well-being of individuals with ADs and a history of cancer.

## Introduction

In the United States, the number of cancer survivors has substantially increased because of advancements in early detection and treatment [1]. In 2022, more than 18 million Americans were living with cancer, accounting for 5.4% of the population, and this figure is projected to increase to 22.5 million by 2032 [1,2]. However, cancer survivors often have long-term symptoms resulting from the disease and its treatment, which may negatively affect their physical functioning, emotional well-being, and overall quality of life [3,4]. In addition, cancer survivors have increased risks of chronic conditions, including infections, endocrine disorders, weight fluctuations, cardiovascular problems, fatigue, sleep problems, and emotional and psychological disturbances [3–5].

Atopic diseases (ADs), including asthma, eczema, food allergies, and allergic rhinitis, are characterized by an overactive immune response to common environmental antigens. These conditions are associated with comorbid chronic health problems, such as a reliance on systemic immunosuppressant medications, impaired sleep [6], malnutrition [7], hypertension and cardiovascular disease [8], inflammation [9,10], and mental health disorders (depression and anxiety) [11,12]. Cancer survivors can be particularly susceptible to ADs. In addition, patients with an atopic condition who have survived cancer may have more severe symptoms than those who have never had cancer because of the immunosuppressive nature of cancer therapies [13].

Cancer treatments, such as chemotherapy and radiation, suppress the immune system. Cancer itself is considered a systemic disease that can induce changes in the immune system, affecting the efficacy of immunotherapies [13]. This immunosuppression may lead to the development or exacerbation of ADs in several ways. First, an altered immune response after treatment might affect the prevalence or severity of atopic symptoms. Second, these treatments might induce changes in the skin or respiratory system, triggering or worsening atopic conditions. Furthermore, the body's inflammatory response to cancer treatment might interact with inflammation typically associated with ADs.

The prevalence of allergic diseases in the general population is 30%–40%, making these diseases a major chronic inflammatory health concern. The World Health Organization has identified allergic diseases as one of the three main disorders to be prioritized for prevention and control in the 21st century [14,15]. Because of continual advancements in cancer treatments, the population of cancer survivors is increasing in size. Thus, the prevalence of ADs among cancer survivors is also expected to rise. Understanding the incidence and specific characteristics of atopic conditions in this

population is essential. Such insights can assist public health officials, clinical practitioners, homecare nurses, and oncology nurses in providing tailored care and health management strategies to cancer survivors. Studies have focused on the relationships among mental health, physical function, falls, and mortality in cancer survivors [3,16–18] and the association between ADs and the risk of certain types of cancer in the general population [19,20]. However, few studies have examined the prevalence of ADs specifically in cancer survivors at a population level. Thus, the present study estimated the prevalence of ADs among cancer survivors by using nationally representative data from the US National Health Interview Survey (NHIS) and compared this prevalence with the prevalence in the general population.

## Materials and methods

### Study design and population

The NHIS, conducted annually by the Centers for Disease Control and Prevention through the National Center for Health Statistics, is a national cross-sectional survey targeting civilian, noninstitutionalized participants in the United States. The survey employs a stratified, multistage sampling design to select approximately 35,000 households from randomly chosen clusters. From each household, a sample of adults aged 18 years or older is randomly selected for detailed interviews on their health status, health-care services, lifestyle risk factors, and other health-related topics. In 2021, the household response rate for the survey was 52.8%. To account for sampling design and nonresponse, rigorous procedures have been implemented. Additional details regarding the design and methodology of the NHIS were published elsewhere [21]. We used data from the 2021 NHIS, which included questions designed to evaluate participants' histories of ADs and cancer (Fig 1A). Participants without a history of ADs, those with unknown cancer history, those with missing responses to the general condition question, and those with missing demographic information were excluded in this study. After the application of these exclusion criteria, the final sample comprised 3651 cancer survivors and 24,966 adults without a history of cancer (Fig 1B). The NHIS design has been reviewed and approved by the Institutional Review Board of the Centers for Disease Control and Prevention. Because the present study involved secondary analyses of publicly available and deidentified NHIS data, no additional institutional review board approval was required. This study adhered to the Strengthening the Reporting of Observational Studies in Epidemiology reporting guidelines.

### Main covariates

Cancer survivors were identified on the basis of participants' responses (yes or no) to the question of whether a doctor or other health professional had ever informed them of a diagnosis of cancer or malignancy of any type. Individuals were classified as cancer survivors from the time of their initial cancer diagnosis to the survey year. The cancer site was determined using responses to a follow-up question regarding the location of the cancer. ADs were defined using two self-reported questions (Fig 1B).

### Other covariates

Sociodemographic characteristics included age, sex (female or male), race/ethnicity (self-reported as Mexican American, non-Hispanic Black, non-Hispanic White, or other, which included other Hispanic, other non-Hispanic, and non-Hispanic multiracial), marital status (categorized as married or living with a partner; divorced, separated, or widowed; and never married), educational level (less than a high school diploma, high school diploma, or more than a high school diploma), health insurance status (covered or uncovered), and the ratio of family income to poverty-level income. The poverty income ratio (PIR) was used to classify income levels as follows: high income (PIR ≥4), middle income (PIR >1 and <4), and income at or below the federal poverty level (PIR ≤1).

Lifestyle risk factors were smoking status (never, former, or current) and body mass index (BMI), calculated as weight in kilograms divided by height in meters squared. The BMI categories were defined as underweight (BMI < 18.5 kg/m2),

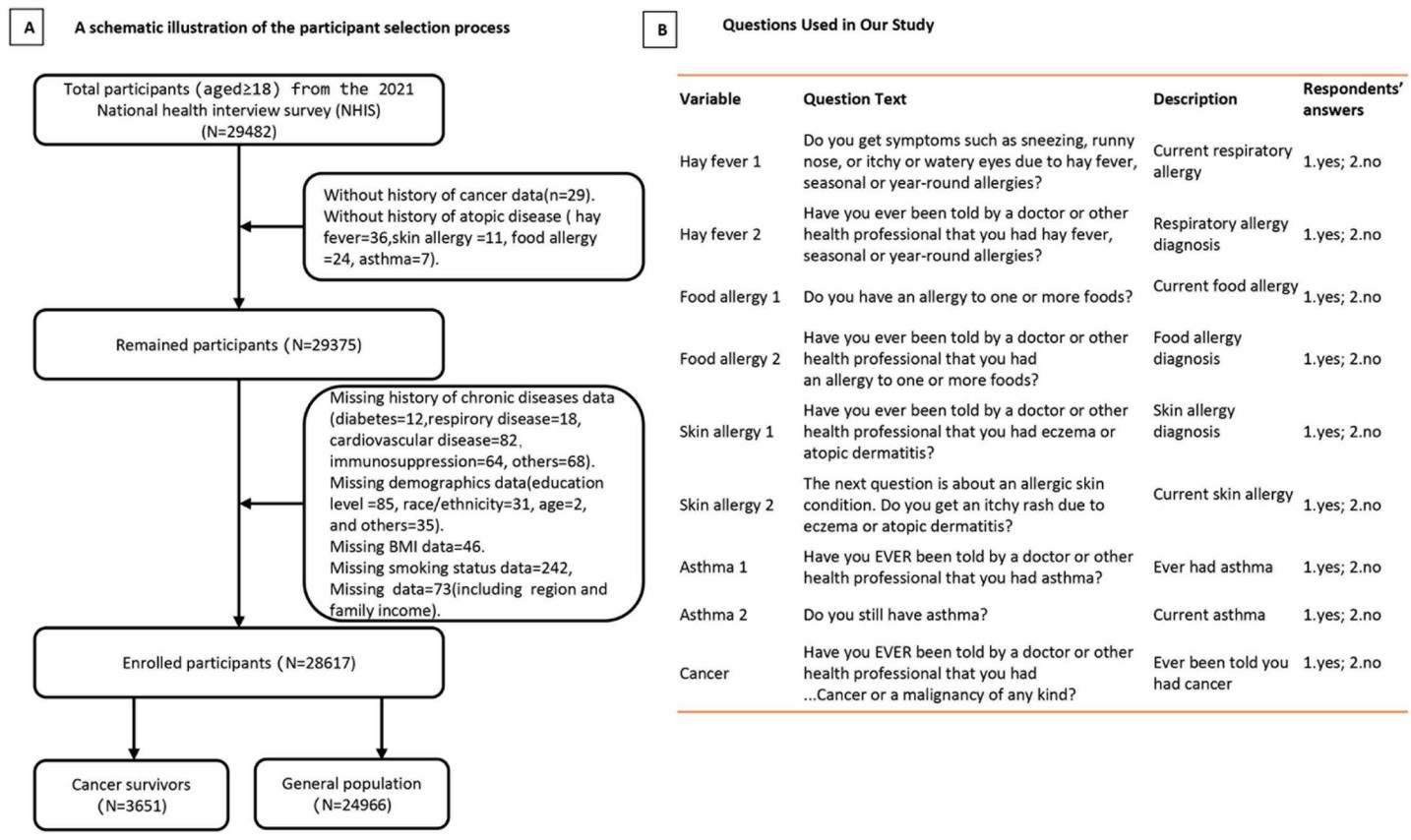

**Fig 1. Flow chart of sample selection from the NHIS 2021.**

healthy weight (BMI = 18.5–24.9 kg/m2), overweight (BMI = 25.0–29.9 kg/m2), and obese (BMI ≥ 30 kg/m2). The number of comorbidities (0, 1–2, or ≥3) was determined on the basis of self-reported chronic medical conditions, including diabetes, hypertension, COVID-19, coronary heart disease, stroke, chronic obstructive pulmonary disease, arthritis, liver disease, and kidney disease. Self-reported health status was assessed using a Likert-like scale with five categories: poor, fair, good, very good, and excellent.

## Statistical analysis

Categorical variables are presented as numbers and percentages, whereas continuous variables are presented as means with standard deviations. The baseline characteristics of cancer survivors and the general population were compared. Differences in these characteristics between the two groups (i.e., effect size) were determined using odds ratios (ORs) and 95% confidence intervals (CIs). These values were calculated using STATA 16 survey frequency procedures for categorical variables. All analyses accounted for sampling bias by incorporating survey weights provided by the National Center for Health Statistics. The unadjusted and weighted prevalence of ADs between the two groups were compared using the chi-square test.

To evaluate the association between cancer survivorship and the prevalence of ADs, multivariable logistic regression models were adjusted for age, sex, race/ethnicity, BMI category, insurance status, family income-to-poverty ratio, education level, geographic region, residential location, smoking status, COVID-19 status, number of COVID-19 vaccinations,

and comorbidity burden. The comorbidities accounted for in the models included a history of diabetes mellitus, a history of hypertension, high cholesterol levels, coronary heart disease, myocardial infarction, congestive heart failure, dementia, stroke, immunosuppression, chronic obstructive pulmonary disease, emphysema, arthritis, liver disease, hearing loss, and kidney disease.

To examine the association between the prevalence of ADs and cancer survivorship, both adjusted and unadjusted logistic regression models were used. For multivariable analysis, variables were incorporated a priori into the fully adjusted model. Subgroup analyses were performed to examine ADs among cancer survivors on the basis of age group, sex, race/ethnicity, and number of comorbidities. In addition, multivariate logistic regression models were used to evaluate the relationship between tumor type and survival time. Stratified analyses were conducted to determine associations by specific cancer types. Statistical analyses were performed using SPSS and Stata. A two-tailed p value of <0.05 was considered statistically significant.

## Results

### Baseline characteristics

Of the 28,617 patients enrolled in this study (weighted n = 245,213,047), 48.3% were women and 51.7% were men. Of all patients, 12.8% were cancer survivors (Table 1). The cancer survivors were more likely to be female (OR, 1.08; 95% CI, 1.06–1.10), aged 65–84 years (OR, 3.18; 95% CI, 3.16–3.19), and aged 85 years or older (OR, 4.00; 95% CI, 3.89–4.13). They were also more likely to be non-Hispanic White (OR, 1.36; 95% CI, 1.35–1.37), live in the western region of the United States (OR, 1.08; 95% CI, 1.07–1.10), reside in a medium and small metropolitan area (OR, 1.13; 95% CI, 1.11–1.14), and live in a nonmetropolitan area (OR, 1.21; 95% CI, 1.20–1.23). Furthermore, the cancer survivors were more likely to have an educational level higher than a high school diploma (OR, 1.08; 95% CI, 1.07–1.09), have a family income-to-poverty ratio greater than 4 (OR, 1.12; 95% CI, 1.10–1.15), be overweight (OR, 1.04; 95% CI, 1.02–1.05), have health insurance coverage (OR, 1.08; 95% CI, 1.07–1.09), and be a former smoker (OR, 1.72; 95% CI, 1.70–1.74). Regarding general health status, the cancer survivors were more likely to report good (OR, 1.21; 95% CI, 1.20–1.23), very good (OR, 1.96; 95% CI, 1.94–1.97), or excellent (OR, 3.04; 95% CI, 3.03–3.05) health compared with the general population. They were 3.08 times more likely (95% CI, 3.06–3.09) to have three or more comorbidities compared with the general population. The prevalence of ADs among the cancer survivors was as follows: 13.6% (95% CI, 10.1%–11.0%) for current skin allergy, 13.4% (95% CI, 12.9%–13.9%) for persistent asthma, 54.5% (95% CI, 52.6%–56.4%) for current hay fever, and 11.2% (95% CI, 10.0%–12.4%) for current food allergy. These rates were higher than those in the general population (Table 1).

### Association between cancer survivorship and the prevalence of ADs

The multivariable analysis of the 2021 NHIS data revealed that having current hay fever (adjusted OR [aOR], 1.14; 95% CI, 1.05–1.23; p < 0.001), persistent asthma (aOR, 1.17; 95% CI, 1.03–1.33; p < 0.001), current skin allergy (aOR, 1.26; 95% CI, 1.13–1.41; p = 0.017), or current food allergy (aOR, 1.22; 95% CI, 1.08–1.38; p = 0.002) were significantly associated with cancer survivorship (Table 2). The number of comorbid ADs was also significantly associated with a history of cancer. In particular, having three ADs was associated with a significantly higher likelihood of being a cancer survivor (aOR, 1.58; 95% CI, 1.30–1.93; p < 0.001). Having two ADs was associated with a moderate increase in this likelihood (aOR, 1.25; 95% CI, 1.11–1.41; p < 0.001), whereas having one AD was associated with a mild increase in this likelihood (aOR, 1.17; 95% CI, 1.08–1.27; p < 0.001; Table 2). However, having all four ADs was not associated with an increased likelihood of being a cancer survivor (aOR, 1.26; 95% CI, 0.79–1.99; p < 0.001).

Subgroup analyses were conducted by stratifying the ADs into specific conditions on the basis of age, sex, race/ethnicity, and number of comorbidities. The cancer survivors were more likely to have current hay fever than were the general population (aOR, 1.11; 95% CI, 1.03–1.20; Fig 2A). This trend was particularly evident among individuals aged 45–64

**Table 1. Baseline characteristics of participants comparing cancer survivors and the general population[a].**

| Characteristic | Total | Cancer survivors | General population | Effect size, OR (95% CI)[b] |
|---|---|---|---|---|
| Total, No. [weighted No.] (%) | 28617[245 213 047] | 3651[24 709 353] (12.8) | 24966[220 503 694](87.2) | NA |
| **Sex, No. (%)** | | | | |
| Male | 48.3(47.6-48.9) | 44.5(42.7-46.4) | 48.7(47.9-49.4) | 0.91(0.89-0.93) |
| Female | 51.7(51.1-52.4) | 55.5(53.6-57.3) | 51.3(50.6-52.1) | 1.08(1.06-1.10) |
| **Age categories,y** | | | | |
| 18-44 | 45.6(44.9-46.3) | 10.0(8.8-11.5) | 49.5(48.8-50.3) | 0.20(0.19-0.21) |
| 45-64 | 32.1(31.5-32.8) | 30.9(29.1-32.7) | 32.3(31.6-33.0) | 0.96(0.92-1.01) |
| 65-84 | 19.9(19.4-20.4) | 51.8(50.0-53.7) | 16.3(15.9-16.8) | 3.18(3.16-3.19) |
| ≥85 | 2.4(2.2-2.6) | 7.2(6.4-8.1) | 1.8(1.7-2.0) | 4.00(3.89-4.13) |
| **Race/ethnicity** | | | | |
| American Hispanic | 9.7(9.3-10.1) | 2.8(2.2-3.6) | 10.5(10.0-11.0) | 0.27(0.26-0.28) |
| Other Hispanic | 6.6(6.3-7.0) | 3.3(2.6-4.3) | 7.0(6.6-7.4) | 0.47(0.46-0.48) |
| Non-Hispanic White | 63.4(62.7-64.1) | 83.3(81.7-84.7) | 61.2(60.5-61.9) | 1.36(1.35-1.37) |
| Non-Hispanic Black | 11.7(11.2-12.2) | 6.6(5.7-7.6) | 12.2(11.7-12.8) | 0.54(0.50-0.58) |
| Non-Hispanic Asian | 5.9(5.6-6.2) | 2.1(1.6-2.7) | 6.3(6.0-6.7) | 0.33(0.32-0.35) |
| Other/Methnicity | 2.7(2.5-3.0) | 1.9(1.5-2.6) | 2.8(2.7-3.1) | 0.68(0.66-0.70) |
| **Region** | | | | |
| West | 17.3(16.8-17.9) | 18.4(17.0-20.0) | 17.2(16.6-17.8) | 1.08(1.07-1.10) |
| Midwest | 21.0(20.4-21.5) | 20.8(19.4-22.3) | 21.0(20.4-21.6) | 0.99(0.96-1.01) |
| Northeast | 38.1(37.4-38.4) | 38.2(36.3-40.0) | 38.1(37.4-38.8) | 1.00(0.98-1.02) |
| South | 23.6(23.0-24.2) | 22.6(21.1-24.2) | 23.7(23.1-24.3) | 0.95(0.93-0.96) |
| **Urban-Rural classification** | | | | |
| Large central metro | 31.6(30.9-32.2) | 26.1(24.4-27.9) | 32.2(31.5-32.9) | 0.81(0.78-0.84) |
| Large fringe metro | 24.0(23.4-24.6) | 23.5(21.9-25.1) | 24.1(23.5-24.7) | 0.98(0.95-1.01) |
| Medium and small metro | 30.9(30.3-31.6) | 34.4(32.7-36.2) | 30.5(29.9-31.2) | 1.13(1.11-1.14) |
| Nonmetropolitan | 13.5(13.0-13.9) | 16.0(14.7-17.4) | 13.2(12.7-13.7) | 1.21(1.20-1.23) |
| **Educational level** | | | | |
| Less than high school | 9.3(8.9-9.7) | 8.2(7.2-9.4) | 32.2(31.5-32.9) | 0.25(0.24-0.26) |
| High school graduate or GED | 28.2(27.6-28.9) | 25.0(23.4-26.7) | 24.1(23.5-24.7) | 1.04(0.99-1.10) |
| More than high school | 62.5(61.8-63.2) | 66.8(64.9-68.5) | 62.0(61.3-62.7) | 1.08(1.07-1.09) |
| **BMI categories** | | | | |
| Underweight | 1.8(1.6-2.0) | 2.6(2.1-3.3) | 1.7(1.5-1.9) | 1.53(1.52-1.54) |
| Normal weight | 31.6(31.0-32.3) | 31.3(29.6-33.0) | 31.7(31.0-32.3) | 0.99(0.97-1.01) |
| Overweight | 34.1(33.5-34.8) | 35.5(33.7-37.3) | 34.0(33.3-34.7) | 1.04(1.02-1.05) |
| Obese | 32.5(31.8-33.1) | 30.7(28.9-32.4) | 32.7(32.0-33.4) | 0.94(0.92-0.95) |
| **Family income-to-poverty ratio** | | | | |
| <1 | 9.8(9.4-10.2) | 6.9(6.1-7.9) | 10.1(9.9-10.6) | 0.68(0.67-0.69) |
| 1.00-1.99 | 17.4(16.8-17.9) | 15.7(14.4-17.2) | 17.5(17.0-18.1) | 0.90(0.88-0.91) |
| 2.00-3.99 | 29.5(28.8-30.1) | 29.4(27.7-31.1) | 29.5(28.8-30.1) | 1.00(0.98-1.01) |
| ≥4 | 43.4(42.7-44.1) | 48.0(46.1-49.9) | 42.8(42.1-43.6) | 1.12(1.10-1.15) |
| **Smoking status** | | | | |
| Never | 65.1(64.4-65.7) | 51.7(49.8-53.6) | 66.6(65.9-67.3) | 0.78(0.75-0.79) |
| Former | 23.1(22.5-23.6) | 37.0(35.2-38.8) | 21.5(21.0-22.1) | 1.72(1.70-1.74) |
| Current | 11.8(11.4-12.3) | 11.4(10.2-12.7) | 11.9(11.4-21.4) | 0.96(0.94-0.97) |

*(Continued)*

**Table 1.** (Continued)

| Characteristic | Total | Cancer survivors | General population | Effect size, OR (95% CI)[b] |
|---|---|---|---|---|
| Total, No. [weighted No.] (%) | 28617[245 213 047] | 3651[24 709 353] (12.8) | 24966[220 503 694](87.2) | NA |
| **No. of comorbidities[d]** | | | | |
| 0 | 37.0(36.3-37.7) | 12.6(11.3-14.0) | 39.7(39.0-40.4) | 0.33(0.32-0.34) |
| 1 | 21.9(21.3-22.5) | 18.1(16.8-19.6) | 22.3(21.7-22.9) | 0.81(0.80-0.83) |
| 2 | 16.5(16.0-17.0) | 19.6(18.2-22.0) | 21.8(21.2-22.4) | 0.90(0.89-0.91) |
| ≥ 3 | 24.6(24.0-25.2) | 49.6(47.8-51.5) | 16.1(15.6-16.7) | 3.08(3.06-3.09) |
| **Insurance status** | | | | |
| Yes | 91.5(91.1-91.9) | 97.7(96.9-98.3) | 90.8(90.3-91.3) | 1.08(1.07-1.09) |
| No | 8.5(8.1-8.9) | 2.3(1.7-3.1) | 9.1(8.7-9.7) | 0.25(0.25-0.26) |
| **General health status** | | | | |
| Poor | 24.7(24.1-25.3) | 11.2(10.1-12.4) | 26.2(25.5-26.8) | 0.43(0.42-0.44) |
| Fair | 34.2(33.6-34.9) | 29.5(27.8-31.3) | 34.7(34.0-35.4) | 0.85(0.83-0.87) |
| Good | 27.8(27.1-28.4) | 33.1(31.3-34.9) | 27.2(26.5-27.8) | 1.21(1.20-1.23) |
| Very good | 10.4(10.0-10.8) | 18.6(17.2-20.1) | 9.5(9.0-10.0) | 1.96(1.94-1.97) |
| Excellent | 3.0(2.8-3.2) | 7.6(6.7-8.7) | 2.5(2.3-2.7) | 3.04(3.03-3.05) |
| **Having atopic disease by a specialist** | | | | |
| Current hay fever | 49.5(48.8-50.2) | 54.5(52.6-56.4) | 48.9(48.2-19.7) | 1.13(1.11-1.14) |
| Current skin allergy | 10.9(10.5-11.3) | 13.6(10.1-15.0) | 10.6(10.1-11.0) | 1.28(1.26-1.29) |
| Current food allergy | 9.9(9.5-10.3) | 11.2(10.0-12.4) | 9.7(9.3-10.2) | 1.15(1.14-1.17) |
| Still asthma | 13.5(13.1-14.0) | 14.7(13.4-16.2) | 13.4(12.9-13.9) | 1.10(1.08-1.11) |

Abbreviations: NA, not applicable; OR, odds ratio.

[a]Data from the 2021 year cycle NHIS.

[b]Unadjusted OR using the general population as the reference group. Effect size provides estimates of differences between the cancer survivor and general population groups.

[c]Body mass index [BMI; calculated as weight in kilograms divided by height in meters squared) was calculated using self-reported weight and height (underweight, <18.5; normal weight, 18.5–24.9; overweight, 25.0–29.9; and obese ≥30.0)]

[d]Self-reported comorbidities included diabetes mellitus, hypertension, COVID-19, coronary heart disease, myocardial infarction, congestive heart failure, epilepsy, dementia, stroke, immunosuppression, dyslipidemia, disability, chronic obstructive pulmonary disease, emphysema, hepatitis, arthritis, chronic fatigue syndrome, liver disease, and kidney disease.

years (aOR, 1.19; 95% CI, 1.04–1.39), women (aOR, 1.24; 95% CI, 1.12–1.37), and the non-Hispanic White population (aOR, 1.10; 95% CI, 1.01–1.20). The difference in the prevalence of current hay fever between cancer survivors and the general population was most pronounced in individuals with a single comorbidity (aOR, 1.27; 95% CI, 1.07–1.52). The cancer survivors had 20% higher odds of current food allergy than did the general population (unadjusted prevalence: 11.1% vs. 9.9%; 95% CI, 1.06–1.36; Fig 2B). This association was particularly significant among individuals aged 18–44 years (aOR, 1.49; 95% CI, 1.07–2.07), those aged 85 years or older (aOR, 1.85; 95% CI, 1.10–3.13), women (aOR, 1.27; 95% CI, 1.10–1.46), and the non-Hispanic White population (aOR, 1.23; 95% CI, 1.07–1.40). The difference in the prevalence of food allergy between the cancer survivors and general population was largest in individuals with three or more comorbidities (aOR, 1.23; 95% CI, 1.07–1.48). The cancer survivors had 12% higher odds of current skin allergy than did the general population (aOR, 1.12; 95% CI, 1.01–1.26; Fig 2C). Similar patterns were discovered in individuals aged 18–44 years (aOR, 1.22; 95% CI, 1.15–1.28), those aged 45–64 years (aOR, 1.28; 95% CI, 1.05–1.55), those aged 65–84 years (aOR, 1.23; 95% CI, 1.05–1.49), women (aOR, 1.21; 95% CI, 1.06–1.39), the non-Hispanic Asian population (aOR, 2.22; 95% CI, 1.19–4.15), and the non-Hispanic White population (aOR, 1.16; 95% CI, 1.03–1.31). The cancer survivors had 15 times higher odds of persistent asthma than did the general population (95% CI, 1.01–1.31; Fig 2D).

**Table 2. Association between risk of atopic disease and cancer survivors in the NHIS, 2021.**

| | General population | | Cancer Survivors | | Crude OR(95% CI) | P Value | Adjusted OR(95% CI)[a] | P Value |
|---|---|---|---|---|---|---|---|---|
| | Frequency | Prevalence (95% CI) | Frequency | Prevalence (95% CI) | | | | |
| **Current hay fever** | | | | | | | | |
| No | 12617 | 88.3(87.8-88.8) | 1677 | 11.7(11.2-12.3) | 1[Reference] | | 1[Reference] | |
| Yes | 12349 | 86.2(85.6-86.8) | 1976 | 13.8(13.2-14.4) | 1.20(1.12-1.29) | <0.001 | 1.14(1.05-1.23) | <0.001 |
| **Current skin allergy** | | | | | | | | |
| No | 22291 | 86.4(86.0-86.8) | 3518 | 13.6(13.2-14.0) | 1[Reference] | | 1[Reference] | |
| Yes | 2675 | 84.4(83.2-85.7) | 493 | 15.6(14.3-16.8) | 1.13(1.17-1.44) | <0.001 | 1.26(1.13-1.41) | <0.001 |
| **Still asthma** | | | | | | | | |
| No | 22915 | 87.3(86.9-87.7) | 3290 | 12.5(12.1-12.9) | 1[Reference] | | 1[Reference] | |
| Yes | 2051 | 85.0(83.6-86.5) | 361 | 15.0(13.5-16.4) | 1.23(1.09-1.38) | 0.001 | 1.16(1.07-1.35) | 0.017 |
| **Current food allergy** | | | | | | | | |
| No | 22482 | 87.4(87.0-87.7) | 3247 | 12.6(12.2-13.0) | 1[Reference] | | 1[Reference] | |
| Yes | 2484 | 86.0(84.7-87.3) | 404 | 14.0(12.7-15.3) | 1.17(1.01-1.26) | 0.037 | 1.22(1.08-1.38) | 0.002 |
| **Atopic Disease, No.** | | | | | | | | |
| 0 | 10862 | 88.8(88.2-89.3) | 1376 | 11.2(10.7-11.8) | 1[Reference] | | 1[Reference] | |
| 1 | 9776 | 86.5(85.9-87.1) | 1526 | 13.5(12.9-14.1) | 1.23(1.14-1.33) | <0.001 | 1.17(1.08-1.27) | <0.001 |
| 2 | 3361 | 85.6(85.4-86.7) | 566 | 14.4(13.3-15.5) | 1.33(1.20-1.48) | <0.001 | 1.25(1.11-1.41) | <0.001 |
| 3 | 807 | 83.6(81.3-86.0) | 158 | 16.4(14.0-18.7) | 1.55(1.29-1.85) | <0.001 | 1.58(1.30-1.93) | <0.001 |
| 4 | 160 | 86.5(81.6-91.4) | 25 | 13.5(8.58-18.4) | 1.23(0.81-1.89) | 0.334 | 1.26(0.79-1.99) | 0.321 |

Abbreviations: NHIS, National Health Interview Survey; OR, odds ratio.

Binary logistic regression models were constructed with Cancer Survivors as the binary dependent variable and history of hay fever, skin allergy, asthma, food allergy, and the number of atopic diseases as the binary independent variable. Multivariable models were constructed that included age, sex, race/ethnicity, annual household income, educational level, insurance coverage, Covid-19 and number of COVID-19 vaccinations, and self-reported comorbidities including diabetes mellitus, hypertension, coronary heart disease, myocardial infarction, congestive heart failure, epilepsy, dementia, stroke, immunosuppression, dyslipidemia, disability, chronic obstructive pulmonary disease, emphysema, hepatitis, arthritis, chronic fatigue syndrome, liver disease, and kidney disease. In addition, multivariable models of hay fever, skin allergy, asthma, and food allergy included each of the other atopic disorders. Crude and adjusted ORs and 95% CIs were reported.

This pattern was also found among individuals aged 45–64 years (aOR, 1.34; 95% CI, 1.08–1.66) and the non-Hispanic White population (aOR, 1.16; 95% CI, 1.03–1.31). When restricted to cancer survivors and stratified by time since cancer diagnosis, those who had received their cancer diagnosis more than 5 years earlier had higher odds of current hay fever (aOR, 1.18; 95% CI, 1.01–1.38) than did those who had received their cancer diagnosis within 3 years (aOR, 0.97; 95% CI, 0.75–1.23) or 3–5 years (reference group). Similarly, those who had received their cancer diagnosis more than 5 years earlier had higher odds of current skin allergy (aOR, 1.39; 95% CI, 1.05–1.83) than did those who had received their cancer diagnosis within 3 years (aOR, 10.7; 95% CI, 0.70–1.65) or 3–5 years (reference group). However, no associations were discovered between persistent asthma or current food allergy and the time since cancer diagnosis (Table 3).

## Association between cancer type and AD prevalence

Analysis of the 2021 NHIS data confirmed an association between certain types of cancer and ADs. After adjusting for various confounding factors in multivariable models, we found higher odds of current hay fever in individuals with head and neck cancer (aOR, 1.32; 95% CI, 1.05–1.60), skin cancer (aOR, 1.35; 95% CI, 1.20–1.53), or gynecologic cancer (aOR, 1.43; 95% CI, 1.23–1.80). Furthermore, we noted increased odds of current skin allergy in individuals with head and neck cancer (aOR, 1.39; 95% CI, 1.02–1.78), skin cancer (aOR, 1.35; 95% CI, 1.33–1.86), or gynecologic cancer

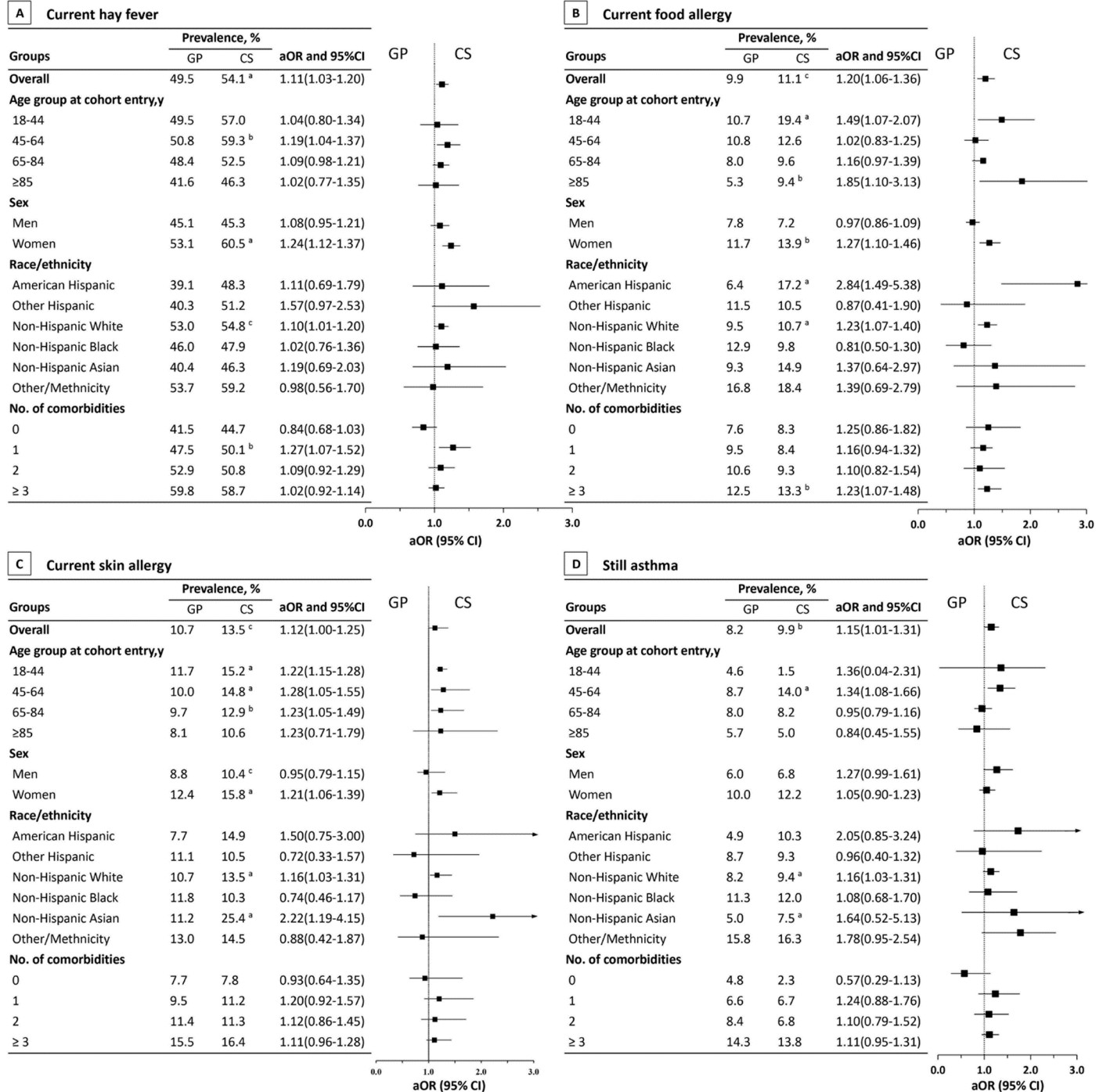

**Fig 2. Atopic Diseases Comparing Cancer Survivors (CS) With the General Population (GP).** Crude weighted prevalence for atopic disease, including current hay fever **(A)**, current food allergy **(B)**, current skin allergy **(C)**, and still asthma **(D)**. aOR indicates an adjusted odds ratio. CI: confidence interval. [a] P < 0.001. [b] P < 0.01. [c] P <0.05.

(aOR, 1.38; 95% CI, 1.03–1.83). Similarly, the odds of persistent asthma were higher in those with head and neck cancer (aOR, 1.59; 95% CI, 1.02–2.49), gynecologic cancer (aOR, 1.69; 95% CI, 1.02–2.59), digestive system cancer (aOR, 1.53; 95% CI, 1.05–2.24), and lung cancer (aOR, 1.79; 95% CI, 1.06–3.03). Increased odds of current food allergy were identified in individuals with head and neck cancer (aOR, 1.94; 95% CI, 1.30–2.91) or gynecologic cancer (aOR, 1.62; 95% CI, 1.22–2.14). Other cancer types were not associated with AD prevalence among cancer survivors.

## Discussion

The present study is the first to examine the prevalence of ADs in cancer survivors by using a large, population-based sample from the NHIS. By using a nationally representative sample of US adults aged 18 years and older, we examined the prevalence of self-reported current hay fever, current skin allergy, persistent asthma, and current food allergy among cancer survivors and compared these findings with those for the general population. The analysis revealed significantly higher prevalence of current hay fever, current skin allergy, persistent asthma, and current food allergy in the cancer survivors than in the general population. These differences persisted even after adjustments were made for potential risk factors. In addition, our findings confirmed variations in the prevalence of AD by sex, race, and age among cancer survivors.

Our study found that women, older adults, and non-Hispanic White individuals were more susceptible to ADs. The prevalence of ADs among cancer survivors appears to vary by age, sex, and race, likely due to various factors, including immune system changes, long-term treatment effects, age-related physiological alterations, genetic predisposition, environmental exposures, lifestyle choices, socioeconomic status, and access to health care. Additional research is necessary to determine these differences and develop more personalized treatment strategies tailored to patients on the basis of their sex, age, and racial or ethnic background [22–24].

The current study provides valuable insights into the associations between cancer types and the prevalence of ADs among cancer survivors. Increased odds of ADs were discovered in individuals with head and neck cancer, gynecologic cancer, digestive system cancer, or lung cancer. Functional impairments resulting from head and neck tumor surgeries can include disruptions to salivary gland function, chewing, oral health, and swallowing. These impairments can negatively affect overall health, both physiologically and psychologically. Impaired upper digestive tract function is associated with decreased masticatory ability, limited food selection, and poor nutrition [25]. For example, individuals with incomplete dentition may opt to swallow food without adequate chewing [25], which can lead to malnutrition, decreased immune function, and complications in anticancer treatments [26]. Furthermore, head and neck tumor surgeries can alter facial appearance, impair speech and respiration, cause visual and auditory dysfunction, and reduce self-confidence, self-image, and social activities, collectively affecting quality of life [27,28]. Because cancer survivors encounter long-term illnesses and psychological stress, the additional burden of ADs may exacerbate their physical discomfort and emotional challenges. The mechanisms through which patients with head and neck tumors develop ADs after treatment may involve multiple aspects [29,30]. First, cancer treatments, such as radiotherapy and chemotherapy, can suppress immune function, increasing sensitivity to allergens. For example, radiotherapy may impair salivary gland function, resulting in dry mouth and altered oral microbiota, which can increase susceptibility to fungal infections, including Candida, and subsequently trigger allergic diseases. Chemotherapy can cause mucositis, weakening the skin and mucosal barriers, and increasing the likelihood of allergic reactions. Second, changes in nutritional status, including weight loss and malnutrition due to treatment, can compromise immune function and increase the prevalence of ADs. Furthermore, certain medications, such as antibiotics and other therapeutic agents, may act as allergens, triggering allergic responses. In addition, patients undergoing head and neck tumor treatments may use innovative therapies, such as immune checkpoint inhibitors, which can affect immune responses and occasionally lead to immune-related side effects, including ADs.

Digestive system cancer treatments can disrupt the gut microbiota, leading to dysbiosis, which is closely related to the occurrence and progression of ADs. The mechanisms underlying this relationship include the following aspects [31–34]: (1) The gut microbiota regulates immune function by promoting the development of the intestinal immune system and

**Table 3. Unadjusted unweighted prevalence and AOR among cancer survivors stratified by cancer type[a].**

| | Crude prevalence of Cancer survivors % (95% CI) | aOR (95% CI)[b] |
|---|---|---|
| **Still asthma[n = 2412; general population(%), 8.2(7.9–8.6)]** | | |
| Head and neck cancers(n=169)[c] | 14.4(9.1-19.7) | 1.59(1.02-2.49) |
| Skin cancer(n=1276)[d] | 9.2(7.7-10.8) | 1.22(0.99-1.50) |
| Gynecologic cancers(n=374)[e] | 15.6(11.7-19.4) | 1.69(1.02-2.59) |
| Blood cancers(n=199)[f] | 8.5(4.7-12.4) | 0.97(0.58-1.61) |
| Digestive cancers(n=262)[g] | 12.6(8.6-16.6) | 1.53(1.05-2.24) |
| Breast cancer(N=697) | 8.3(6.3-10.4) | 1.12(0.89-1.41) |
| Brain cancer(n=23) | 17.4(1.9-32.9) | 1.93(0.64-5.85) |
| Bone cancer(n=30) | 13.3(11.7-25.5) | 1.49(0.50-4.37) |
| Prostate cancer(n=455) | 6.4(4.1-9.6) | 0.82(0.54-1.24) |
| Lung cancer(N=115) | 15.7(9.0-22.3) | 1.79(1.06-3.03) |
| Urinary tract cancers(n=121)[h] | 8.3(3.4-13.2) | 1.09(0.56-2.11) |
| Melanoma cancer(n=189)[i] | 10.1(5.8-14.3) | 1.32(0.81-2.15) |
| Other cancers(n=310)[j] | 11.9(8.3-15.5) | 1.34(0.94-1.93) |
| **Current skin allergy[n = 3168; general population(%), 10.7(10.3–11.1)]** | | |
| Head and neck cancers(n=169)[c] | 15.6(10.1-21.1) | 1.39(1.01-1.78) |
| Skin cancer(n=1276)[d] | 15.4(13.4-17.3) | 1.57(1.33-1.86) |
| Gynecologic cancers(n=374)[e] | 17.9(13.8-21.9) | 1.38(1.03-1.83) |
| Blood cancers(n=199)[f] | 12.6(8.0-17.2) | 1.13(0.74-1.73) |
| Digestive cancers(n=262)[g] | 10.7(6.9-14.4) | 0.98(0.65-1.46) |
| Breast cancer(N=697) | 13.5(11.0-16.0) | 1.20(0.95-1.51) |
| Brain cancer(n=23) | 4.3(-4.0-12.7) | 0.33(0.04-2.47) |
| Bone cancer(n=30) | NA | NA |
| Prostate cancer(n=455) | 9.5(6.8-12.1) | 0.92(0.65-1.29) |
| Lung cancer(N=115) | 7.0(2.3-11.6) | 0.53(0.23-1.09) |
| Urinary tract cancers(n=121)[h] | 14.0(7.9-20.2) | 1.31(0.78-2.22) |
| Melanoma cancer(n=189)[i] | 11.1(6.9-15.6) | 1.06(0.67-1.68) |
| Other cancers(n=310) [j] | 13.9(10.0-17.7) | 1.20(0.86-1.67) |
| **Current hay fever[n = 14323;general population(%), 49.5(48.8–50.1)]** | | |
| Head and neck cancers(n=169)[c] | 59.3(51.8-66.7) | 1.31(1.05-1.60) |
| Skin cancer(n=1276)[d] | 58.3(55.6-61.0) | 1.35(1.20-1.53) |
| Gynecologic cancers(n=374)[e] | 66.3(61.3-71.3) | 1.43(1.13-1.80) |
| Blood cancers(n=199)[f] | 49.2(42.2-56.2) | 0.90(0.68-1.20) |
| Digestive cancers(n=262)[g] | 45.0(39.0-51.1) | 0.81(0.63-1.04) |
| Breast cancer(N=697) | 56.5(52.8-60.2) | 1.09(0.93-1.28) |
| Brain cancer(n=23) | 34.8(15.3-54.2) | 0.47(0.19-1.14) |
| Bone cancer(n=30) | 49.5(22.5-57.5) | 0.59(0.28-1.25) |
| Prostate cancer(n=455) | 41.8(37.2-46.3) | 0.84(0.69-1.03) |
| Lung cancer(N=115) | 50.4(41.3-59.6) | 0.94(0.64-1.37) |
| Urinary tract cancers(n=121)[h] | 48.8(39.9-57.7) | 0.95(0.66-1.37) |
| Melanoma cancer(n=189)[i] | 54.0(46.9-61.1) | 1.15(0.89-1.55) |
| Other cancers(n=310)[j] | 52.9(47.3-58.5) | 0.98(0.78-1.24) |
| **Food allergy[n = 2888; general population(%), 9.9(9.6–10.3)]** | | |
| Head and neck cancers(n=169)[c] | 18.0(12.1-23.8) | 1.94(1.30-2.91) |
| Skin cancer(n=1276)[d] | 10.6(8.9-12.3) | 1.19(0.98-1.45) |

*(Continued)*

**Table 3.** (Continued)

| | Crude prevalence of Cancer survivors % (95% CI) | aOR (95% CI)[b] |
|---|---|---|
| Gynecologic cancers(n=374)[e] | 18.7(14.6-22.8) | 1.62(1.22-2.14) |
| Blood cancers(n=199)[f] | 8.0(4.2-11.8) | 0.83(0.49-1.39) |
| Digestive cancers(n=262)[g] | 11.1(7.3-14.9) | 0.81(0.63-1.04) |
| Breast cancer(N=697) | 12.8(10.2-15.2) | 1.21(0.95-1.53) |
| Brain cancer(n=23) | 8.9(-2.8-20.2) | 0.82(0.19-3.56) |
| Bone cancer(n=30) | 13.3(11.7-25.5) | 1.51(0.52-4.43) |
| Prostate cancer(n=455) | 7.7(5.2-10.1) | 1.14(0.78-1.66) |
| Lung cancer(N=115) | 4.3(0.6-8.1) | 0.48(0.19-1.20) |
| Urinary tract cancers(n=121) | 5.0(1.1-8.8) | 0.60(0.26-1.37) |
| Melanoma cancer(n=189) | 6.9(3.3-10.5) | 0.75(0.43-1.33) |
| Other cancers(n=310)[h] | 8.4(5.3-11.5) | 0.84(0.56-1.27) |

Abbreviations: aOR, adjusted odds ratio; NA, not applicable.

All calculations indicate unweighted numbers.

[b]Model adjusted for age, sex, race/ethnicity, region, educational level, family income-to-poverty ratio, insurance status,urban-rural classification, BMI, smoking status, COVID-19 and number of COVID-19 vaccinations, and the number of comorbidity burden, using the prevalence of general population as the reference.

[c]Head and neck cancers included cancers of the mouth, tongue, lip, larynx, trachea, throat or pharynx, pharyngeal, and thyroid gland.

[d]Skin cancer included skin non-melanoma cancer, Skin cancer (don't know what kind), and basal or squamous cells of the skin.

[e]Gynecologic cancers cervical cancer, uterine cancer, and ovarian cancer.

[f]Blood cancers included leukemia, lymphoma, and myeloma.

[g]Digestive cancers included cancers of the rectum, colon, esophagus, gallbladder, liver, pancreas, and stomach.

[h]Up to three different kinds of cancers could be mentioned by the sample adult. This is a cancer type that indicates that other kinds of cancer, not included in previous cancer kind types, were mentioned. This type also includes kidney and testicular cancer.

inducing T-cell differentiation, which helps maintain immune balance. Allergic diseases are associated with incomplete immune system development and impaired immune regulation. The distribution and diversity of the gut microbiota in children with allergies substantially differ from those in healthy children, indicating the critical role of the gut microbiota in immune function. (2) Maintaining a balance between regulatory T cells and helper T cells [17] is crucial for immune homeostasis. In allergic diseases, this balance is disrupted, leading to excessive activation of helper T cells [17] and inflammatory responses. (3) An overly hygienic lifestyle can reduce cancer survivors' exposure to environmental antigens, causing a shift in the immune response toward Th2 pathways and increasing the risk of allergic diseases. (4) Damage to the intestinal barrier and dysbiosis can facilitate the translocation of allergens, triggering systemic immune responses and exacerbating allergic conditions. (5) The gut microbiota modifies the bile acids, affecting their bioavailability and bioactivity. Changes in bile acid metabolism have been associated with the development of allergic diseases because of their role in the host's metabolic reactions. (6) The gut microbiota also regulates immune and metabolic processes by modulating the activity of aryl hydrocarbon receptors.

A study reported that ADs are associated with iron deficiency anemia [35]. Cancer can lead to iron deficiency anemia through several mechanisms. Cancer cells consume substantial amounts of iron to support their growth, resulting in depletion of iron stores within the body. In addition, cancer treatments, including chemotherapy and radiation therapy, can cause anemia by impairing the bone marrow's ability to produce red blood cells. Tumor treatments may also result in dyspepsia and nutrient absorption dysfunction [36–38]. Iron deficiency anemia can manifest as a secondary effect of food avoidance and malnutrition after anticancer treatment. Diets lacking milk products and other essential nutrients can cause malnutrition [39]. Studies have demonstrated a relationship between ADs and malnutrition, indicating that individuals with

ADs have a higher risk of vitamin D deficiency [40–41]. Iron deficiency can lead to fatigue, impaired small-bowel function, stunted growth, and hindered cognitive development. Moreover, iron deficiency is associated with decreased attention span and intelligence, impaired sensory perception, and behavioral and emotional changes [36,38,41]. Restrictive diets often adopted by individuals with suspected food allergies or those experiencing exacerbation of skin or airway conditions because of specific foods may further contribute to the malnutrition observed in individuals with ADs.

Cancer and its treatments can have extensive physical and emotional effects on survivors. Immediate side effects may include fatigue, pain, lymphedema, neuropathy, bone or joint problems, and sexual dysfunction. Survivors may also experience late-onset effects, such as secondary cancers, organ damage, and chronic conditions. In addition, survivors have increased risks of heart and digestive problems, hormonal imbalances, cognitive impairments, and psychological challenges, including anxiety and depression. Other potential complications include weight fluctuations caused by changes in metabolism or appetite, a weakened immune system leading to increased susceptibility to infections, oral health problems such as dry mouth and tooth decay, and sleep problems including insomnia [3–5,42].

The relationship between ADs and systemic comorbidities is complex and multifaceted, with potential implications for patient management and public health strategies. ADs are associated with chronic inflammation, which can both exacerbate and be exacerbated by various systemic conditions, including cardiovascular disease, metabolic syndrome, and autoimmune disorders. The immune dysregulation characteristic of ADs can result in altered immune responses, potentially increasing susceptibility to infections and contributing to the development of autoimmune diseases [6–12]. Further research is essential to elucidate the mechanisms underlying the associations between ADs and systemic comorbidities. Such insights could facilitate the development of novel therapeutic approaches.

Cancer survivors may reduce their physical activity for various reasons, including the side effects of cancer treatment, a decline in physical function, psychological challenges, pain and discomfort caused by the cancer itself, and a lack of motivation and support. Survivors may lack the motivation to engage in physical activity or the necessary support from their family, friends, or medical team. Other contributing factors include limited health awareness, insufficient knowledge regarding the benefits of physical activity, and additional health problems such as heart disease or diabetes [43]. Frailty—a medical syndrome characterized by decreased strength, endurance, and physiological function—further increases vulnerability to external stressors and may lead to a decline in physical activity. The underlying causes of frailty in cancer survivors include treatment side effects, the direct effect of cancer, psychological distress, decreased physical capacity, immune-related adverse events after immunotherapy, late toxic effects of treatment, unhealthy lifestyle habits, cancer-related cognitive impairment, fertility and sexual dysfunction, and an increased risk of second primary tumors. Frailty may also be associated with a decline in immune function, potentially increasing susceptibility to allergic reactions from various stimuli. Decreased physical activity may affect the development and progression of ADs through various mechanisms, such as an imbalance in differentiation of T helper cell types 1 and 2, impaired epithelial barrier function, environmental factor interactions, and altered T-cell receptor signal strength. These interconnected factors can collectively contribute to the onset and exacerbation of AD [44–46].

The exact pathophysiological mechanisms underlying the high prevalence of ADs among cancer survivors remain unclear. However, several potential mechanisms have been proposed to explain this association (Fig 3).

## Public health implications

In the present study, the prevalence of current hay fever, current skin allergy, persistent asthma, and current food allergy in the United States was significantly higher in cancer survivors than in the general population at 54.5%, 13.6%, 14.7%, and 8.9%, respectively. The cancer survivors were found to have a three times higher prevalence than the general population of having all three of the four ADs. The prevalence of ADs among cancer survivors varied on the basis of factors such as age, race, sex, and cancer type. In addition, the prevalence of hay fever and skin allergy was discovered to increase as the survival time increased.

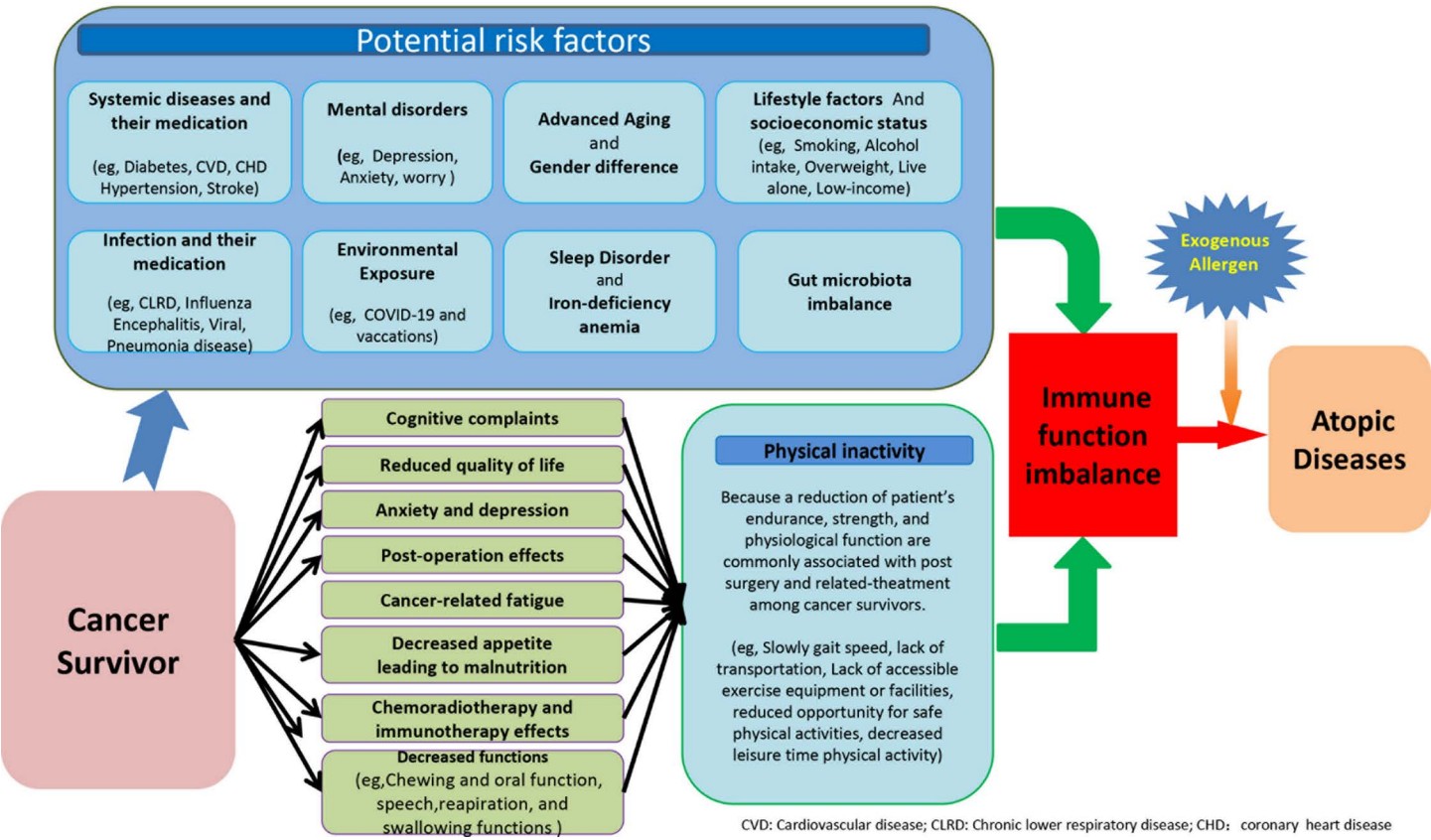

**Fig 3. Schematic Representation of Some Potential Risk Factors and Direct and Indirect Pathways increased the incidence of AD Among Cancer Survivors.**

Research into the characteristics of ADs in cancer survivors has crucial public health implications. Because of industrialization, environmental changes, dietary shifts, and increased exposure to allergens, the global prevalence of ADs is rising, creating a substantial public health challenge. Cancer survivors, who are often exposed to additional allergens through medications and radiation during treatment, may have an increased prevalence of allergic conditions. The co-occurrence of ADs in this population can complicate their health status, adversely affect quality of life, increase treatment complexity, prolong recovery, and substantially contribute to the societal medical burden. Addressing the comorbidity of ADs in cancer survivors and understanding the underlying mechanisms are crucial for developing targeted prevention and treatment strategies. Investigating the prevalence and characteristics of ADs in cancer survivors can enable public health professionals to design tailored health policies and interventions to address the unique needs of this demographic. This includes enhancing public education on AD management, improving health-care providers' expertise, and advocating for systemic reforms to support the well-being and quality of life of cancer survivors. In addition, examining the features of ADs in cancer survivors can provide critical data for shaping public health initiatives. Such efforts can help reduce the personal and societal effects of these conditions and enhance overall public health standards. Furthermore, this research can provide novel insights into the mechanisms underlying ADs in cancer survivors, potentially leading to innovative prevention and treatment strategies.

## Strengths and limitations

This study has several strengths. First, it utilized a nationally representative sample, enabling a comprehensive assessment of AD prevalence among cancer survivors. Moreover, the study benefited from the detailed

**Table 4. AD (crude prevalence [%] and 95%CI) among cancer survivors, by time since cancer diagnosis.**

| Time since cancer diagnosis | ≤ 3 years n=940(weighted N= 3465470; 14.0%) | | >3, ≤ 5 years n=396(weighted N= 2810100; 11.4%) | | > 5 years n=2315 (weighted N= 15116815; 61.2%) | | p for trend |
|---|---|---|---|---|---|---|---|
| | prevalence%, 95%CI | aOR,95%CI[a] | prevalence%, 95%CI | aOR,95%CI[a] | prevalence%, 95%CI | aOR,95%CI[a] | |
| Current hay fever | 50.9(47.7-54.0) | Reference | 51.0(46.1-56.0) | 0.97(0.75-1.23) | 55.9(53.9-57.9) | 1.18(1.01–1.38)[b] | 0.041 |
| Current skin allergy | 11.8(9.7-13.9) | Reference | 13.1(9.8-16.5) | 1.07(0.70-1.65) | 14.3(12.5-15.7) | 1.39(1.05–1.83)[b] | 0.043 |
| Still asthma | 10.1(8.2-12.0) | Reference | 8.3(5.6-11.1) | 0.69(0.39-1.14) | 10.1(8.8-11.3) | 1.10(0.84-1.49) | 0.134 |
| Current food allergy | 11.0(9.0-13.0) | Reference | 11.9(8.7-15.1) | 1.05(0.67-1.64) | 11.0(9.7-12.2) | 1.03(0.76-1.38) | 0.997 |

Abbreviations: AD: atopic disease.CI: confidence interval; aOR: adjusted odds ratio.

[a]Model adjusted for age, sex, race/ethnicity, region, educational level, family income-to-poverty ratio, insurance status,urban-rural classification, BMI, smoking status, COVID-19 and number of COVID-19 vaccinations, and the number of comorbidity burden.

[b]p< 0.05.

and high-quality data provided by the NHIS, which enabled the control of potential confounding effects across a wide range of demographic, socioeconomic, chronic disease, health status, and lifestyle factors. Another notable strength is the large sample size of cancer survivors and individuals with ADs, providing robust statistical power. Furthermore, the study incorporated data on the specific time interval from cancer diagnosis and cancer type (Table 4).

This study has several limitations. First, its cross-sectional design prevented the discovery of cause-and-effect relationships. Second, the history of ADs was based on self-reports instead of diagnostic tests, which may have led to either underestimation or overestimation of their prevalence due to their episodic nature. Third, the reliance on self-reported data introduced the potential for misclassification and recall bias. Fourth, the limited availability of tumor- and therapy-related data—such as diagnosis stage, histological type, and treatment methods—hindered a definitive understanding of the etiology of ADs. Fifth, the generalizability of the study's findings is restricted to adult patients with cancer in the United States and does not extend to other populations, including patients with cancer and ADs. Finally, the absence of physical activity data limited our ability to evaluate such activity's role in preventing allergic diseases in cancer survivors. Given these limitations, future research should involve larger cohorts, employ case–control designs, and incorporate comprehensive diagnostic assessments to validate and expand upon the current findings.

## Conclusions

This cross-sectional analysis revealed that a significantly higher prevalence of ADs in cancer survivors than in the general population. Gaining a deeper understanding of associations can enable public health professionals and researchers to develop more effective strategies and policies. These initiatives can be tailored to address the unique needs of cancer survivors, with the ultimate goal of improving their long-term health outcomes.

## Acknowledgments

This document used data adapted from the Centers for Disease Control and Prevention (CDC).We would like to express our sincere gratitude to the Centers for Disease Control and Prevention (CDC) and the National Health Service (NHS) for their invaluable support and contributions to our research project. We are grateful to all those who participated in the National Health Service (NHS) In addition, we sincerely thank the participants of the database we used in this study.

## Author contributions

**Conceptualization:** Lu Xiao, Yuhui Shen.

**Data curation:** Lu Xiao, Yuhui Shen, Nana xiang, juebo yu.

**Formal analysis:** Lu Xiao, juebo yu.

**Funding acquisition:** Yuhui Shen.

**Investigation:** Lu Xiao, juebo yu.

**Methodology:** Nana xiang, juebo yu.

**Project administration:** juebo yu.

**Resources:** Nana xiang.

**Software:** Lu Xiao, juebo yu.

**Supervision:** Yuhui Shen, Nana xiang, juebo yu.

**Validation:** Nana xiang.

**Writing – original draft:** Lu Xiao, Yuhui Shen, juebo yu.

**Writing – review & editing:** Yuhui Shen, juebo yu.

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
