## [Decision Letter · Decision Letter 0]

26 Dec 2024

PONE-D-24-46805Risk of Atopic Disease Among Adult Cancer Survivors Insights from the 2021 NHISPLOS ONE

Dear Dr., yu,

Thank you for submitting your manuscript to PLOS ONE. After careful consideration, we feel that it has merit but does not fully meet PLOS ONE’s publication criteria as it currently stands. Therefore, we invite you to submit a revised version of the manuscript that addresses the points raised during the review process.

Please read carefully the comments from the reviewers and respond to them point-by-point.If you are unable to provide a response to any of the comments, then give an explanation of the reason for this.You must address the review comments by the academic editor that are restated below.

We look forward to receiving your revised manuscript.

Kind regards,

Benjamin Ansa

Academic Editor

PLOS ONE

Journal Requirements:

2. We note that your Data Availability Statement is currently as follows: If the data are all contained within the manuscript and/or Supporting Information files, enter the following: All relevant data are within the manuscript and its Supporting Information files.

Additional Editor Comments:

The author has presented a manuscript of a study titled "Risk of Atopic Disease Among Adult Cancer Survivors Insights from the 2021 NHIS" for publication consideration. The results of the study reveal that cancer survivors have a significantly higher prevalence of atopic disease than the general population.

There are comments below for author consideration to help improve the quality of the paper.

1.Abstract: The sample size mentioned here is 28,617. This is different from that in the methods and in Figure 1 (2,069+18,369=20438). Revise this.

2. Methods: Include more information about the 2021 NHIS in the paper. What are the response rates, the validity and reliability indices of the survey instrument? Mention the specific exclusion criteria in the methods section as well.

Statistical analysis: The use of prevalence and odds ratio for measuring risk is inappropriate. Rather, the attack rate and relative risk are the appropriate measures for measuring risk. Rate Ratios (RR) or Rate Differences (RD) may also be used for calculating the risk differences between the comparison groups.

If you decide not to reanalyze the data, then I suggest you change the title and objectives to "exploring the prevalence of Atopic Disease Among Adult Cancer Survivors: Insights from the 2021 NHIS".

3. Results: The results provided in the manuscript are not reflective of the measures for RISK. This should be revised per my suggestion in the Methods or change the narrative to the measures of prevalence of AD among cancer survivors.

The tables and figures look okay.

4. Conclusions: The narrative here states "prevalence of AD in cancer survivors". However, the title and objectives mention RISK. Be consistent. Otherwise, the explanation for the relationship between atopic disease and cancer was well presented in the conclusions.

5. References: Look okay.

Minor comments

1. Avoid using the phrase "To our knowledge". This was used several times in the manuscript and should be deleted.

2. You may solicit the services of a scientific English writing to review the paper.

Best of Luck!

Reviewers' comments:

Reviewer's Responses to Questions

**Comments to the Author**

1. Is the manuscript technically sound, and do the data support the conclusions?

Reviewer #1: Partly

2. Has the statistical analysis been performed appropriately and rigorously? 

Reviewer #1: I Don't Know

3. Have the authors made all data underlying the findings in their manuscript fully available?

Reviewer #1: Yes

4. Is the manuscript presented in an intelligible fashion and written in standard English?

Reviewer #1: No

5. Review Comments to the Author

Reviewer #1: The manuscript is based on the original idea of using the publicly available data from the annual nhis report for a statistical study. It must be accepted that the data collected is based solely on the information provided by the respondents and cannot be objectively verified. This considerably reduces the scientific significance of such an approach. The approach of looking for a link between cancer and atopic diseases is interesting. However, i would suggest to the individual author that, if the manuscript is revised, other co-authors with expertise in their respective fields should be consulted, namely a statistician, an allergologist and, if possible, an epidemiologist. This would strengthen the classification of the results and the credibility of the statistical analysis. The manuscript also has considerable linguistic and formal deficiencies, which would have to be revised before a new submission.

6. PLOS authors have the option to publish the peer review history of their article (what does this mean? ). If published, this will include your full peer review and any attached files.

**Do you want your identity to be public for this peer review?** For information about this choice, including consent withdrawal, please see our Privacy Policy .

Reviewer #1: No

---

## [Author Response · Author response to Decision Letter 1]

15 Jan 2025

Dear editors and reviewers:

On behalf of all the contributing authors, I would like to express our sincere appreciations of your letter and reviewers’ constructive comments concerning our article entitled Exploring the prevalence of Atopic Disease Among Adult Cancer Survivors: Insights from the 2021 NHIS. These comments are all valuable and helpful for improving our article. According to the reviewers’ comments, we have made extensive modifications to our manuscript to make our results convincing. In this revised version, changes to our manuscript were all highlighted within the document by using itemized text. Point-by-point responses to the nice associate editor and the nice reviewers are listed below this letter.

Response to Editor Comments:

Question 1: .Abstract: The sample size mentioned here is 28,617. This is different from that in the methods and in Figure 1 (2,069+18,369=20438). Revise this.

Response 1 : Thank you for your comments on my manuscript. We have checked and corrected it. After the application of these exclusion criteria, the final sample comprised 3651 cancer survivors and 24,966 adults without a history of cancer (Fig 1B).

Question 2: Methods: Include more information about the 2021 NHIS in the paper. What are the response rates, the validity and reliability indices of the survey instrument? Mention the specific exclusion criteria in the methods section as well.

Statistical analysis: The use of prevalence and odds ratio for measuring risk is inappropriate. Rather, the attack rate and relative risk are the appropriate measures for measuring risk. Rate Ratios (RR) or Rate Differences (RD) may also be used for calculating the risk differences between the comparison groups.

If you decide not to reanalyze the data, then I suggest you change the title and objectives to "exploring the prevalence of Atopic Disease Among Adult Cancer Survivors: Insights from the 2021 NHIS".

.

Response 2: Thank you for your comments on my manuscript. The household response rate for 2021 stood at 52.8%. To account for the sample design and nonresponse, extensive procedures have been implemented. Participants without a history of ADs, those with unknown cancer history, those with missing responses to the general condition question, and those with missing demographic information were excluded in this study.

The questionnaire tool used in this study is a survey tool designed by NHlS database in the United States. Although we cannot obtain accurate the validity and reliability indices of the survey instrument . But I believe this specific one is reliable and effective. Otherwise, this questionnaire could not be run in this institution for nearly ten years. NHIS's survey questionnaire tools are carefully designed and continuously optimized to effectively collect high-quality health data, providing important data support for health research and public health decision-making the validity and reliability indices of the survey instrument. NHIS's survey questionnaire tools can be downloaded from the internet

(https://ftp.cdc.gov/pub/Health_Statistics/NCHS/Dataset_Documentation/NHIS/2021/adult-codebook.pdf)

Question 3: Results: The results provided in the manuscript are not reflective of the measures for RISK. This should be revised per my suggestion in the Methods or change the narrative to the measures of prevalence of AD among cancer survivors.

The tables and figures look okay.

Response 3: Thank you for your comments on my manuscript.We have checked and corrected it.

Question 4: Conclusions: The narrative here states "prevalence of AD in cancer survivors". However, the title and objectives mention RISK. Be consistent. Otherwise, the explanation for the relationship between atopic disease and cancer was well presented in the conclusions.

Response 4: Thank you for your comments on my manuscript.We have checked and corrected it.

Question 5: References: Look okay.

Response 5: Thank you for your comments on my manuscript.

Question 6: Minor comments

1. Avoid using the phrase "To our knowledge". This was used several times in the manuscript and should be deleted.

2. You may solicit the services of a scientific English writing to review the paper.

Response 6: Thanks for your suggestion.

1. We have checked and corrected it.

2. I read through the full text and correct these errors. In addition, we feel sorry for our poor writings, however, we do invite a professional English editing agency which is a native English speaker from the USA to polish our article. And we hope the revised manuscript could be acceptable for you. I hope you can give me some good advice again. I will revise it seriously.

Thank you very much your attention and time. Look forward to hearing from you.

Response to Reviewer #1: The manuscript is based on the original idea of using the publicly available data from the annual nhis report for a statistical study. It must be accepted that the data collected is based solely on the information provided by the respondents and cannot be objectively verified. This considerably reduces the scientific significance of such an approach. The approach of looking for a link between cancer and atopic diseases is interesting. However, i would suggest to the individual author that, if the manuscript is revised, other co-authors with expertise in their respective fields should be consulted, namely a statistician, an allergologist and, if possible, an epidemiologist. This would strengthen the classification of the results and the credibility of the statistical analysis. The manuscript also has considerable linguistic and formal deficiencies, which would have to be revised before a new submission.

Question of Reviewer #1:

Dear Reviewer, thank you for your thorough evaluation of the manuscript and for highlighting the areas that require improvement. Your insights are invaluable in enhancing the quality and credibility of the research. Here is how we plan to address each of your concerns: We acknowledge the limitations inherent in relying on self-reported data from the National Health Interview Survey(NHIS).To mitigate this, we will include a detailed discussion on the potential biases and limitations of our data source in the revised manuscript. Additionally, we will explore the possibility of corroborating our findings with other studies or datasets that might offer more objective measures. Your suggestion to involve co-authors with specific expertise is well-taken. We seek collaboration with a statistician to ensure the robustness of our statistical analysis. We also consult with an allergologist to provide clinical insights into the link between cancer and atopic diseases. We invite an epidemiologist to strengthen our understanding of the population-level implications of our findings. We revise the manuscript to include a more detailed and rigorous statistical analysis. This involve refining our methods and ensuring that the statistical tests used are appropriate for the data and the research questions posed. We believe that that the results are presented in a manner that is clear and understandable to readers. We are committed to improving the manuscript's language and presentation. We engage a professional editor to review and polish the manuscript for clarity, coherence, and adherence to academic standards. This includes correcting grammatical errors, improving the flow of the text, and ensuring that the manuscript conforms to the journal's formatting guidelines. We appreciate your constructive feedback and are dedicated to revising the manuscript to meet the high standards of scientific research and publication. We believe that with these revisions, the manuscript will provide a more substantial contribution to the field. Thank you once again for your valuable comments.

Yours sincerely,

Juebo Yu

13 June 2023

Department of Otolaryngology, Affiliated Hospital of Yangzhou University, Yangzhou university, Yangzhou, China

Corresponding Author: Juebo Yu, MD, Department of Otolaryngology, Affiliated Hospital of Yangzhou University, No. 45 Taizhou Rd, Guangling District, Yangzhou 225001, China (yujuebo2004@163.com).

---

## [Editor Report · Decision Letter 1]

19 Jan 2025

PONE-D-24-46805R1Exploring the prevalence of atopic disease among adult cancer survivors:insights from the 2021 NHISPLOS ONE

Dear Dr. yu,

Thank you for submitting your manuscript to PLOS ONE. After careful consideration, we feel that it has merit but does not fully meet PLOS ONE’s publication criteria as it currently stands. Therefore, we invite you to submit a revised version of the manuscript that addresses the points raised during the review process.

**See below the reviewer's comment and an attached file for the minor revision of your paper.**

We look forward to receiving your revised manuscript.

Kind regards,

Benjamin Ansa

Academic Editor

PLOS ONE

Journal Requirements:

Additional Editor Comments:

Dear Dr. juebo yu,

Thank you for submitting the revised version of your manuscript titled "Exploring the prevalence of atopic disease among adult cancer survivors: insights from the 2021 NHIS". You have responded to most of the reviewers' comments.

However, there are still minor revisions to be made before the publication consideration for your paper can move forward.

There are still parts of the manuscript that refer to your analysis as "Risk of ADs" instead of "Prevalence of AD's". This should be corrected. I have highlighted in yellow color those areas to be corrected in the attached file for your convenience.

We hope you can resubmit the revised version soon.

---

## [Author Response · Author response to Decision Letter 2]

11 Mar 2025

Dear editors and reviewers:

On behalf of all the contributing authors, I would like to express our sincere appreciations of your letter and reviewers’ constructive comments concerning our article entitled Exploring the prevalence of Atopic Disease Among Adult Cancer Survivors: Insights from the 2021 NHIS. These comments are all valuable and helpful for improving our article. According to the reviewers’ comments, we have made extensive modifications to our manuscript to make our results convincing. In this revised version, changes to our manuscript were all highlighted within the document by using itemized text. Point-by-point responses to the nice associate editor is listed below this letter.

Response to editor Comments:

Question 1: .Thank you for submitting the revised version of your manuscript titled "Exploring the prevalence of atopic disease among adult cancer survivors: insights from the 2021 NHIS". You have responded to most of the reviewers' comments.

However, there are still minor revisions to be made before the publication consideration for your paper can move forward.

There are still parts of the manuscript that refer to your analysis as "Risk of ADs" instead of "Prevalence of AD's". This should be corrected. I have highlighted in yellow color those areas to be corrected in the attached file for your convenience.

We hope you can resubmit the revised version soon.

Response 1 : Thank you for your comments on my manuscript. We have checked and corrected it..We hope the revised manuscript could be acceptable for you. I hope you can give me some good advice again. I will revise it seriously.

Thank you very much your attention and time. Look forward to hearing from you.

Yours sincerely,

Juebo Yu

13 June 2023

Department of Otolaryngology, Affiliated Hospital of Yangzhou University, Yangzhou university, Yangzhou, China

Corresponding Author: Juebo Yu, MD, Department of Otolaryngology, Affiliated Hospital of Yangzhou University, No. 45 Taizhou Rd, Guangling District, Yangzhou 225001, China (yujuebo2004@163.com).

---

## [Editor Report · Decision Letter 2]

24 Mar 2025

Exploring the prevalence of atopic disease among adult cancer survivors: insights from the 2021 NHIS

PONE-D-24-46805R2

Dear Dr. Yu,

We’re pleased to inform you that your manuscript has been judged scientifically suitable for publication and will be formally accepted for publication once it meets all outstanding technical requirements.

Kind regards,

Benjamin Ansa

Academic Editor

PLOS ONE

Additional Editor Comments (optional):

The author has revised the manuscript according to the review comments.

---

## [Editor Report · Acceptance letter]

PONE-D-24-46805R2

PLOS ONE

Dear Dr. yu,

I'm pleased to inform you that your manuscript has been deemed suitable for publication in PLOS ONE. Congratulations! Your manuscript is now being handed over to our production team.

Kind regards,

on behalf of

Dr. Benjamin Ansa

Academic Editor

PLOS ONE